# Beyond Quiescent and Active: Intermediate Microglial Transcriptomic States in a Mouse Model of Down Syndrome

**DOI:** 10.3390/ijms25063289

**Published:** 2024-03-14

**Authors:** Álvaro Fernández-Blanco, Cèsar Sierra, Clara Tejido, Mara Dierssen

**Affiliations:** 1Centre for Genomic Regulation (CRG), The Barcelona Institute of Science and Technology, 08003 Barcelona, Spain; alvaro.fernandez@crg.eu; 2Laboratory of Neuroepigenetics, Brain Mind Institute, School of Life Sciences, École Polytechnique Fédérale de Lausanne (EPFL), 1015 Lausanne, Switzerland; cesar.sierra@epfl.ch; 3Neuroimmunology and Brain Tumor Immunology (D170), German Cancer Research Center, 69120 Heidelberg, Germany; 4Human Pharmacology and Clinical Neurosciences Research Group, Neurosciences Research Program, Hospital del Mar Medical Research Institute (IMIM), 08003 Barcelona, Spain; 5Universitat Pompeu Fabra (UPF), 08003 Barcelona, Spain; 6Centro de Investigación Biomédica en Red de Enfermedades Raras (CIBERER), 28029 Madrid, Spain

**Keywords:** Down syndrome, microglia, disease-associated microglia

## Abstract

Research on microglia in Down syndrome (DS) has shown that microglial activation, increased inflammatory gene expression, and oxidative stress occur at different ages in DS brains. However, most studies resulted in simplistic definitions of microglia as quiescent or active, ignoring potential intermediate states. Indeed, recent work on microglial cells in young DS brains indicated that those evolve through different intermediate activation phenotypes before reaching a fully activated state. Here we used single nucleus RNA sequencing, to examine how trisomy affects microglial states in the Ts65Dn mouse model of DS. Despite no substantial changes in the proportion of glial populations, differential expression analysis revealed cell type-specific gene expression changes, most notably in astroglia, microglia, and oligodendroglia. Focusing on microglia, we identified differential expression of genes associated with different microglial states, including disease-associated microglia (DAMs), activated response microglia (ARMs), and human Alzheimer’s disease microglia (HAMs), in trisomic microglia. Furthermore, pseudotime analysis reveals a unique reactivity profile in Ts65Dn microglia, with fewer in a homeostatic state and more in an intermediate aberrantly reactive state than in euploid microglia. This comprehensive understanding of microglial transcriptional dynamics sheds light on potential pathogenetic mechanisms but also possible avenues for therapy for neurodevelopmental disorders.

## 1. Introduction

Down syndrome (DS) is the most common genetic form of intellectual disability resulting from an extra copy of chromosome 21 and is associated with a spectrum of cognitive impairments. Until very recently, it was assumed that brain phenotypes in DS were accounted for by alterations in the neuronal population [1,2]. However, recent studies have underscored the involvement of microglia, the resident immune cells of the central nervous system (CNS), not only in neuronal alterations but also in learning and memory deficits in DS [3].

Under physiological conditions, microglial cells exist in a surveillant state [4,5] contributing to brain homeostasis due to their involvement in the phagocytosis of cellular debris, misfolded proteins, and apoptotic cells, among others [6]. Microglia also survey and safeguard different neuronal functions through the microglia-neuron crosstalk [7]. However, upon injury or pathogen invasion, they undergo a process of reactivity to different signals characterized by changes in their shape, mobility, and phagocytic activity, which are sustained by the expression of inflammatory-related genes and proteins [8]. Reactive microglia present an amoeboid shape and have phagocytic functions [9]. Nevertheless, this oversimplified dichotomization of “resting or homeostatic” or “activated or reactive” microglial states is a subject of intense debate since it does not fully grasp the intricacy of microglial responses within the framework of different neurodegenerative disorders [5,10]. Recent developments in single-cell technologies have revealed multiple microglial states related to specific developmental, aging, and disease processes [11]. These studies emphasize that the extent of microglial reactivity goes beyond traditional phenotypes, revealing the coexistence of multiple intermediate states concurrently. Consequently, there exists a diverse array of context-dependent microglial states, which vary across species and models. Importantly, each microglial state is associated with specific functions in the brain and is influenced by various modifying factors, such as age, sex, local signals, or peripheral signals. Some of these states are referred to as Disease-Associated Microglia (DAM), and they are linked to various neurological disorders [12,13,14,15,16,17]. 

However, when examining DS, the dichotomous perspective on microglial states persists, despite the evolving understanding in the microglial research field. The current knowledge, coming from multiple studies in the brains of individuals with DS is that trisomic microglia is in a reactive state that contributes to neuroinflammation. Amoeboid and ramified microglia, associated with reactive microglia, has been identified in DS fetuses as early as 17–22 gestational weeks across different brain regions [18], and microglial dystrophy has also been reported in adults with DS [19,20,21]. Furthermore, microglia from children with DS display an increased somatic size compared to euploid controls [20]. Immunoreactivity of microglia to IL-1β, a pro-inflammatory cytokine, was reported in fetuses, neonates, children, and adults with DS, which would suggest chronic neuroinflammation [20]. Consistent with these findings, a mouse model of DS exhibited an increased production of IL-1β and superoxide in microglial cells at fetal stages [22]. Microglial cells in DS also present a transcriptomic signature related to Alzheimer’s disease (AD) and aging characterized by the expression of C1q-complement-related genes [23]. Corroborating these results, adults with DS show increased expression of CD-64 and CD-86, markers of microglial reactivity [24]. Microglia reactivity has also been documented in mouse models of DS, including Ts65Dn [25,26,27,28] and Dp(16)1Yey [3]. Notably, depleting microglia in young Dp(16)1Yey DS mouse model rescue cognition and improved neuronal spine density and activity [3], and anti-inflammatory therapies restore microglial homeostasis and mitigate cognitive deficits in mouse models of DS [3,26,28]. Collectively, these findings suggest that persistent microglial reactivity may contribute to the cognitive defects in DS. 

Utilizing single-nucleus RNA sequencing, we here provide an unbiased examination of gene expression patterns of the main glial cell types in the trisomic hippocampus of Ts65Dn mice at postnatal day 56 (P56), with a focus on trisomic microglia. We chose Ts65Dn for the study because it recapitulates most of the traits identified in DS. Ts65Dn mice, on the other hand, carry a triplication of 43 coding genes that are not homologous to HSA21 and are not triplicated in human DS [29,30]. We selected P56 to be able to compare our results with different databases and previously published studies because at this age the brain is fully developed. In this study, we explored the impact of the trisomy on the microglial transcriptomic profile and whether the trisomy affects the different microglial states within the hippocampus of Ts65Dn mice. 

In conclusion, our study provides a new scenario in which, rather than considering that there is simply a higher number of activated microglia in DS, there is a higher proportion of microglial cells in an aberrant reactive state that might contribute to the neuroinflammation that is reported in DS.

## 2. Results

### 2.1. Unbiased Identification of Different Glial Subtypes in WT and Ts65Dn Hippocampus

Prior research in individuals with DS and in mouse models of DS has extensively documented that trisomy leads to a global disruption of gene expression, affecting not only neurons but also other cell types in the brain [31,32,33,34]. To investigate the different microglial states present in the trisomic hippocampus, we conducted a single-nucleus RNA sequencing experiment in P56 euploid and Ts65Dn littermates. We sequenced a total of 17910 WT and 13633 Ts65Dn nuclei from the NeuN-negative cell population of two euploid and two Ts65Dn mice using 10x technology (Figure 1A; Appendix A).

Cell types were annotated using the R package SingleR (version 1.0.6), showing, that 56% were categorized as oligodendroglia, 16% as astroglia, and 8% as microglia, while the remaining 20% were classified as endothelial cells, ependymal cells, and neurons, which resulted from the insufficient labeling by the NeuN antibody, and other minor cell types (Figure 1B–D, Appendix A). There were no significant differences in cell proportions between the WT and Ts65Dn mice (Figure 1E). Subsequent analyses were restricted to glial cell types.

### 2.2. Trisomic Glial Cells Exhibit Cell-Type-Specific Transcriptomic Alterations

Differential expression analysis of euploid and trisomic glial subtypes (oligodendroglia, astroglia, and microglia) showed a total of 1434 differentially expressed genes (DEGs, Appendix A). Although upregulated genes were enriched in mouse chromosome 16 (Mmu16), a portion of which is triplicated in Ts65Dn, most of the DEGs were distributed across all the genome, reflecting a general transcriptomic perturbation in the trisomic hippocampus, as reported in previous studies [35] (Figure 2A). As also occurs in the neuronal population [35], we found that the impact of the trisomy on the transcriptome is highly cell-type specific (Figure 2B,C; Appendix A), being microglia the cell type with higher DEGs when compared with the number of sequenced cells (Appendix A). In accordance with previous studies [34], we found that in astroglia, most of the DEGs were significantly upregulated, including triplicated genes such as *Dyrk1A*, *App*, *Son*, and *Scaf4* (Figure 2B,C; Appendix A). Instead, both in microglia and oligodendroglia, we mainly found downregulated genes (Figure 2B,C; Appendix A). We identified 532 DEGs between trisomic and euploid microglia (Appendix A) of which 514 out of 532 were predominantly downregulated (Figure 2B,C; Appendix A). *Lgals9*, and *Cd86* part of the microglial sensome, *Spi1*, or *Ilr7*, which are all associated with the regulation of the immune system were significantly and specifically downregulated in trisomic microglia [36]. 

The cell type-specific alteration of the transcriptome was also reflected in the specific molecular pathways affected in each glial type (Figure 2D). Cell migration emerged as a shared pathway enriched across all glial clusters, featuring specific genes associated with neurodevelopmental cell migration such as *Sema6d*, which was upregulated in trisomic astroglia and downregulated in oligodendroglia (Appendix A). Additionally, microglia displayed distinctive pathway enrichments linked to immune system development, endocytosis, myeloid cell differentiation, and signaling involved in the regulation of immune responses. 

### 2.3. More than 60% of Trisomic Microglial Cells Are in an Intermediate Reactive State

To further investigate the alterations in the microglial population of the trisomic hippocampus, we focused on this cluster. This allowed us to identify two main microglial clusters (Figure 3A), enriched either in markers of homeostatic microglia (Cluster 0) or in activation markers (Cluster 1) (Appendix A). Interestingly, the Uniform Manifold Approximation and Projection (UMAP) of this population showed clear differences between genotypes in both clusters 0 and 1 (Figure 3B). First, we observed a shift in the two-dimensional embedding of the homeostatic cluster, with trisomic cells clustering closer to the reactive microglia population (Figure 3B). At the same time, the cluster of reactive microglia tended to be less abundant than in WT counterparts, although the difference was not significant (Appendix A). These differences were not explained by classical microglial activation markers such as *Aif1* (encoding for Iba1; Figure 3C) or *Cd68* (Appendix A). In accordance with these results, we found similar numbers of Iba1+ and CD-68+ cells in brain sections of the hippocampus of euploid and Ts65Dn mice (Appendix A).

These results suggested an alteration in the reactive states in trisomic microglia possibly indicating intermediate states of activation in DS. To obtain insight into the activation trajectories of trisomic and WT microglia, we performed a pseudotime analysis, which ordered microglial cells according to their degree of activation (Figure 3D,E), accounting for the expression of activation genes such as those associated with the major histocompatibility complex II (MHC-II; Appendix A). This analysis revealed striking differences between genotypes. While WT microglia have a pseudotime distribution in which a large proportion of microglial cells are in a resting state and a smaller fraction is activated, most trisomic microglia (67%) lay in an in-between state according to the pseudotime score (Figure 3D,E; Appendix A). These results suggest that trisomic microglia are permanently in an aberrantly reactive state. 

In this study, we also expected DAM state linked to the trisomy in Ts65Dn. Analyzing previously published datasets [37,38], we found 10 DAM genes that were substantially differentially expressed in trisomic microglia and that overlapped with DAM markers, including *Csmd3*, the most downregulated gene in trisomic microglia, or *Cables1*, (Figure 3G, Table 1). 

## 3. Discussion

Both in DS and in mouse models of DS, there is an exacerbated microglial reactivity [3,23,39] that is accompanied by an increased level of pro-inflammatory molecules [40]. However, most studies simply analyze “resting versus activated” and “M1 versus M2” states. This dualistic classification of good or bad microglia is inconsistent with the wide repertoire of microglial states and functions in development, plasticity, aging, and diseases that were elucidated in recent years. Thus, a more in-depth understanding of microglial activation states is needed, acknowledging the existence of intermediate states is essential for advancing our understanding of the role of microglia in DS. Using single nuclei RNA sequencing we investigated the influence of trisomy on the microglial transcriptomic profile and explored whether the trisomy has an impact on different microglial states within the hippocampus of Ts65Dn mice. We identified five main clusters representing different cell types, including astrocytes, oligodendroglia, microglia, endothelial cells, and neurons. 

Differential expression analysis of glial subtypes uncovered a total of 1434 DEGs, with the triplicated region of chromosome 16 (Mmu16) showing a higher number of significantly upregulated DEGs in Ts65Dn mice. The cell-type specificity of the transcriptomic impact on the transcriptome was striking, with mostly upregulated DEGs in astroglia and mostly downregulated in both microglia and oligodendroglia. Interestingly, a similar cell type-specific impact of the trisomy was recently reported in a single-nucleus sequencing study using the Dp16 model [32]. In accordance with our data, a recent study using DS human transcriptome from the dorsolateral prefrontal cortex and cerebellar cortex showed that astroglia-associated genes and oligodendroglia-associated genes were up- and down-regulated, respectively [41], whereas microglial genes were mainly upregulated. These differences might be explained not only by a different cell type-specific transcriptomic profile in those brain regions but also because the study was performed using a DS human brain. However, this study and ours suggest that there is an intricate relationship between gene dosage and cell-type-specific responses to trisomy. 

The identification of cell-type-specific DEGs allowed us to explore the potential biological pathways affected by trisomy. Cell migration emerged as a common pathway enriched in every glial cluster, with specific genes involved in neurodevelopmental cell migration showing altered expression. Moreover, microglia exhibited unique pathway enrichments related to immune system development, endocytosis, myeloid cell differentiation, and immune response regulation signaling which suggests that their function as resident macrophage CNS might be altered. 

We also found a transcriptomic shift comparing euploid and trisomic microglia. Pseudotime analysis revealed a distinctive reactivity profile in Ts65Dn microglia, with a reduced number of trisomic microglia in a homeostatic or surveillant state and a higher proportion of microglia in an intermediate state of reactivity compared to euploid microglia. The pseudotime score correlated with microglial reactivity as illustrated by the activation of the MHC-II.This finding concurs with previous studies showing that steady-stage microglia lack MHC class II, but microglia reactivity is associated with MHC class II induction [42]. However, whether microglia MHC-II acts as antigen presentation for local T-cell activation in the CNS or modulates microglial signaling is a subject of debate [42]. This higher degree of microglial reactivity in individuals with DS and in DS mouse models has been described by different techniques such as morphological [3,26,43] and transcriptomic [32] analysis. Our study refines this general view and proposes that a different and aberrant disease-associated state of trisomic microglia might account for the higher microglial reactivity that is reported in DS and in DS mouse models. 

The techniques used in previous studies might not have been able to capture the different levels or degrees of reactive microglia. As a matter of fact, there is a debate about whether the classical dualistic classification “resting versus activated” should be reviewed and reevaluated according to recent single-cell technologies and multi-omics data. Indeed, utilizing the classical markers of microglia activation such as Iba1 or Cd68 we were not able to identify this intermediate activation state in Ts65Dn microglia. This is probably because microglia in intermediate activation states may exhibit subtle alterations in gene expression that are not easily captured by conventional immunofluorescence markers. However, instead of using a single gene or protein marker to identify microglial states, the combination of different markers and fate-mapping approaches would better capture the population diversity of microglial cells [5]. These studies highlight that the degree of microglial reactivity extends beyond the classical phenotypes and that multiple intermediate states can coexist at the same time [5]. Therefore, a plethora of context-dependent microglial states differ between species and models. Our results suggest that microglial response in DS disease may be more complex than previously anticipated, as evidenced by the discovery of intermediate microglial activation states in Ts65Dn mice. However, the functional significance of these intermediate states is still speculative. The microglial response to neuroinflammation and synaptic dysfunction associated with DS may exhibit transitional stages, which could be represented by intermediate activation levels. Alternatively, they might imply a unique activation phenotype with distinct functional outcomes. 

Recently, scRNA-seq studies have identified different microglial transcriptional signatures typical of disease models such as DAMs [12], microglial neurodegenerative phenotype (MGnD) [13] and activated response microglia (ARMs) [14], human AD microglia (HAMs) [15] and of aging including white matter-associated microglia (WAM) [44] and axon tract-associated microglia (ATMs) [45] among others. Every microglial state is associated with particular functions in the brain. It is known that different modifying factors including, age, sex, local signals, or peripheral signals can influence microglial states [46,47]. Microglia also respond to different neurodevelopmental, neurological, and neurodegenerative disorders by changing their molecular profile, morphology, motility, and function [48,49]. We found 10 genes that were significantly differentially expressed in trisomic microglia that overlapped with DAM signature including *Csmd3*, which was the top downregulated gene in microglia. It was reported that deficiency in the *Csmd3* gene impaired synaptogenesis and neuronal development [50]. Likewise, *Csmd3* deficiency promotes growth retardation, abnormal cortex development, and neurodevelopmental-related phenotypes such as lower body weight and brain size that are accompanied by defective memory in the novel object recognition test, and impaired sociability, among others [50]. Whether the microglial downregulation of *Csmd3* is directly related to neuronal and cognitive alterations will require further studies. However, the transcriptomic deregulation that we found in trisomic microglia extended beyond DAM markers since we also detected a downregulation of *Inpp5d* or *Bin1* [14], which are markers associated with ARMs or reduced expression of *Tnfrsf21* and *Tln2*, which are related to HAMs [15]. Beyond sharing markers with distinct functional microglial states, the transcriptomic signature of trisomic microglia is also characterized by the altered expression of a large number of genes that have not been previously associated with microglial states and that might be specific to DS. This adds a layer of complexity in understanding the cellular and molecular mechanisms by which trisomic microglia might contribute to either maintaining or disturbing brain homeostasis in DS. 

Although the consequences of this transcriptomic shift in the trisomic microglial population are difficult to anticipate, one possible explanation might be that a higher trisomic microglia reactivity state would promote the release of different proinflammatory cytokines such as IL-6 and TNF-α that might contribute to neuroinflammation that is reported in individuals with DS [51,52]. Given that reactive microglia is already detected early during neurodevelopment both in individuals with DS [18,19,20,21] and in DS mouse models [3,22,25,26,27,28], neuroinflammation could be explained by this increased microglial reactivity. In addition to that, it is also known that microglial reactivity increases with aging in DS [53,54], which might also contribute to the progression of DS neuropathology. However, whether microglial reactivity contributes to neuroinflammation or vice versa is not completely understood. As a matter of fact, different studies have shown that restoring microglial homeostasis with different strategies in mouse models of DS was effective in recovering different neuronal alterations, recovering cognitive and behavioral abilities, and even reducing inflammatory markers [3,26].

In conclusion, our study contributes valuable information regarding the cell-type-specific alterations in microglial gene expression associated with trisomy in the Ts65Dn mouse model. The observed changes in transcriptomic profiles and reactivity states provide a foundation for further investigations into the underlying mechanisms and potential therapeutic targets for neurodevelopmental disorders associated with trisomy.

### Limitations

While our study provides valuable insights into the transcriptional dynamics of microglia in the context of DS, there are several limitations that should be taken into consideration In first place, our study relies on the Ts65Dn mouse model of DS, which recapitulates most of the alterations of DS pathology but might not capture the full complexity of the human condition. For this reason, it is advisable to proceed with caution when extrapolating these findings to human DS individuals. Moreover, it is important to recognize that the phenotype observed in Ts65Dn mice could be affected by the presence of triplicated non-HSA21 genes SnRNA-seq is a powerful approach to analyze gene expression at the single-cell level and to delve into transcriptomic alterations within specific cell types. Even so, it is important to remember that this technique might not capture all RNA species since specific low-abundance transcripts might be missed, potentially leading to small biases in the data. Similarly, since we have utilized stringent criteria for cell type identification based on previously established markers, the resolution to distinguish closely related cell populations might be somehow affected. While our study provides valuable insights into the transcriptional dynamics of microglia in DS, findings should be interpreted taking into consideration the aforementioned limitations. Addressing these constraints, in future research studies will be essential to deepening our understanding of the role of microglia in DS and identifying potential therapeutic targets for the treatment of DS-associated neuroinflammation and cognitive impairment. 

## 4. Materials and Methods

### 4.1. Animals

Ts(1716)65Dn (Ts65Dn) mice were obtained through crossings of a B6EiC3Sn a/A-Ts (1716)65Dn (Ts65Dn) female to B6C3F1/J males purchased from The Jackson Laboratory (Bar Harbor, ME, USA). Genotyping was performed by amplifying genomic DNA obtained from the mice’s tail according to the Jackson Laboratory recommended protocol. Mice had access to food and water ad libitum in controlled laboratory conditions with temperature maintained at 22 ± 1 °C and humidity at 55 ± 10% on a 12 h light/dark cycle (lights off 20:00 h). Mice were socially housed in numbers of two to four littermates. The colony of Ts65Dn mice was maintained in the Animal Facilities at the Barcelona Biomedical Research Park (PRBB, Barcelona, Spain).

According to Directive 63/2010 and Member States’ implementation of it, all trials followed the “Three Rs” principle of replacement, reduction, and refinement. The investigation was conducted in accordance with the Standards for Use of Laboratory Animals No. A5388-01 (NIH) and local (Law 32/2007) and European regulations as well as MDS 0040P2 and the Ethics Committee of Parc de Recerca Biomèdica (Comité Ético de Experimentación Animal del PRBB (CEEA-PRBB)). A/ES/05/I-13 and A/ES/05/14 grant the CRG permission to work with genetically modified organisms. See the Ethics section for further information.

### 4.2. Histology

#### 4.2.1. Immunohistochemistry

In order to quantify Iba1 and CD-68 positive cells in the hippocampus, WT, and Ts65Dn mice were transcardially perfused with ice-cold phosphate buffered saline (PBS) followed by 4% paraformaldehyde (PFA) in PBS (pH 7.4). Brains were extracted and post-fixed in 4% PFA at 4 ºC overnight. Brains were then transferred to PBS and 40 μm coronal consecutive brain sections were obtained employing a vibratome (Leica VT1200S, Leica Microsystems, Wetzlar, Germany), collected in PBS and stored in cryoprotective solution (40% PBS, 30%, glycerol and 30% polyethylene glycol) for long-term storage. For immunofluorescence studies, 4–6 sections per mouse were selected according to stereotaxic coordinates Bregma, −1.54 to −2.54 mm, (mouse brain atlas [55]) with the aid of a bright-field microscope (Zeiss Cell Observer HS; Zeiss, Oberkochen, Germany). Brain sections were washed with PBS (3 × 10 min). Then, sections were permeabilized with 0.5% Triton X-100 in PBS (PBS-T 0.5%) (3 × 15 min) and blocked with 10% of Normal Goat Serum (NGS) for 2 h at room temperature (RT). Sections incubated in PBS-T 0.5% and NGS 5% with the primary antibodies overnight at 4 °C washed again (PBS-T 0.5% 3 × 15 min) and incubated with the secondary antibodies (PBS-T 0.5% + NGS 5%) for 2 h at room temperature protected from light. Finally, samples were washed with PBS-T 0.5% (3 × 15 min) followed by PBS washing (3 × 10 min) to remove the detergent and sections were mounted and coverslipped into a pre-cleaned glass slide with Fluoromount-G medium with DAPI (Thermo Fisher Scientific, Waltham, MA, USA #00-4959-52). Iba1 was stained with rabbit anti-Iba1 (1:1000, Wako Chemicals, Neuss, Germany #019-19741) and visualized with anti-rabbit Alexa-647 (1:500; Thermo Fisher Scientific, #A-21443). CD-68 was stained with rat anti-CD-68 (1:2500; Santa Cruz, Dallas, TX, USA #ab53444) and visualized with anti-rat Alexa-488 (1:500; Thermo Fisher Scientific, #A-11006). Prior to immunostaining, an optimization of the primary antibodies and PBS-T conditions was performed. Serial dilutions of primary antibodies ranging from 1:100 to 1:1000 were prepared while maintaining the secondary antibody concentration constant (1:500). By confocal microscopy, the best primary antibody concentration was selected taking into account the achievement of low background noise and the signal level obtained with the same laser configuration.

#### 4.2.2. Cell Counting

In order to quantify the number of Iba1+ cells and Iba1+; CD-68+ cells in the CA1 region, 40 µm coronal sections were taken from the dorsal hippocampus in the coordinates −1.54 to −2.54 mm AP (relative to bregma). Cell densities are expressed as cells/mm^2^. Confocal fluorescence images were acquired on a Leica SP8 scanning laser microscope using a 20X/0.70 NA objective. Cell counting was performed using the Cell Counter plugin on ImageJ software (version 1.8.0_172) (NIH, Bethesda, MD, USA) in a z-stack (3 μm step size). The stratum radiatum CA1 layer was selected as a region of interest (ROI) and was manually delineated according to the DAPI signal in every section.

Alexa 488 and Alexa 647 channels were filtered and combined to produce composite images. Equal cutoff thresholds were applied to remove signal background from images. The number of double-positive (Iba1+ and CD-68) and single-positive (Iba1+) cells were counted in CA1 stratum radiatum in three consecutive sections (spaced 200 μm between them) per mouse. Data were analyzed using R studio. Imaging and quantifications were performed blind to experimental conditions.

### 4.3. Single Nucleus RNA Sequencing

#### 4.3.1. Nucleus Isolation

Mice were sacrificed by cervical dislocation and the hippocampus were dissected and placed in cold Hanks’ Balanced Salt Solution (Sigma, Saint Louis, MO, USA) #55021C). To obtain a nuclei suspension, the “Frankenstein” procedure was used [56]. Each hippocampus was placed in a fresh tube with 500 μL cold EZ lysis buffer (Sigma #3408) and a sterile RNase-free douncer (Mettler Toledo, Columbus, OH, USA #K-749521-1590) was used to homogenize the buffer. To eliminate any leftover material fragments, the homogenate was filtered through a 70 μm-strainer mesh and centrifuged at 500× *g* for 5 min at 4 °C. The nuclei pellet was resuspended in 1.5 mL EZ Lysis Buffer and centrifuged again. The supernatant was discarded and 500 μL of Nuclei Wash and Resuspension Buffer (NWRB, 1X PBS, 1% BSA, and 0.2 U/μL RNase inhibitor (Thermo Scientific #N8080119) was added to the pellet. After incubation, the pellet was resuspended in 1 mL of NWRB. The nuclei suspension was centrifuged once again, and the washing step with 1.5 mL of NWRB was repeated. Nuclei were then resuspended in 500 μL of 1:1000 anti-NeuN antibody conjugated with Alexa Fluor 647 (Abcam, Cambridge, UK) #ab190565) in PBS and incubated in rotation for 15 min at 4 °C. Nuclei were rinsed with 500 μL of NWRB and centrifuged again after incubation. To create a single-nuclei suspension, nuclei were resuspended in NWRB mixed with DAPI and filtered through a 35 μm cell strainer. 

#### 4.3.2. 10x Single-Cell Barcoding, Library Preparation, and Sequencing

NeuN-negative neuronal nuclei were sorted using fluorescent-activated nuclear sorting (FANS). A total of 10,000 nuclei from each sample were sorted directly into a 96-well plate prefilled with 10x RT buffer prepared without the RT Enzyme Mix using a 70 μm nozzle to minimize the volume deposited. Following sorting, RT Enzyme C was added, and the volume of each well was increased to 80 μL with nuclease-free water. The Chromium Single Cell Chip was loaded with 75 μL of the nuclei plus RT mix. The manufacturer’s instructions (10x Genomics Chromium Single Cell Kit Version 3, Pleasanton, CA, USA) were followed for all downstream cDNA synthesis, library preparation, and sequencing. Libraries were sequenced on a NovaSeq 6000 S1 (Illumina, San Diego, CA, USA) to an average depth of approximately 20,000 reads per cell. 

#### 4.3.3. 10x Data Pre-Processing

The readings were matched to the reference genome, including exons and introns, and transformed to mRNA molecule counts using the manufacturer’s CellRanger pipeline (CellRanger v3.0.1). For every nucleus, we quantified the number of genes for which at least one read was mapped, and then discarded any nuclei with fewer than 200 or more than 2500 genes, respectively, to eliminate low-quality nuclei and duplets. Genes found in fewer than six nuclei were discarded. To normalize for differences in coverage, expression values Ei,j for gene I in cell j were calculated by dividing UMI counts for gene I by the sum of UMI counts in nucleus j, multiplying by 10,000 to create TP10K (transcript per 10,000) values, and finally computing log2(TP10K + 1) (using the NormalizeData function from the Seurat package v.2.3.4) [57].

#### 4.3.4. Batch Correction and Scaling Data Matrix

Since samples were processed in two different experiments, batch correction and data scaling were executed as described in [35]. Briefly, Harmony [58] was used on the normalized dataset and then the data were scaled using the ScaleData function from Seurat. The scaled data matrix was then used for dimensionality reduction and clustering. 

### 4.4. Dimensionality Reduction, Clustering and Visualization

Using the RunPCA method in Seurat, we computed the top 60 principle components using the scaled expression matrix restricted to the variable genes. UMAP (Uniform Manifold Approximation and Projection) used the scores from these principal components as input to downstream grouping and visualization (UMAP). The FindNeighbors and FindClusters functions in Seurat (resolution = 0.05) were used to cluster the data. After that, using UMAP, the clusters were visualized. Before integrating with the IntegrateData function, reference anchors between genotypes were found, and the combined data were analyzed using the same procedures.

SingleR (version 1.0.6) [59] was used for single-cell annotation based on the “MouseRNAseqData” dataset from the celldex package (version 1.0.0) and using the “label.main” to assign cell subtypes. Clusters at a resolution of 0.05 were annotated based on the most prevalent predicted cell subtypes. The annotation was further refined by mapping the most enriched genes for each cluster (identified using the FindAllMarkers function) to the cell types of the Linnarson mouse atlas [60].

### 4.5. Identification of Marker Genes within Every Cluster

The FindAllMarkers function was used to find cluster-specific marker genes using a negative binomial distribution (DESeq2). A marker gene was defined as having a detectable expression in >20% of the cells from the related cluster and being >0.25 log-fold greater than the mean expression value in the other clusters. We were able to choose markers that were highly expressed within each cluster while still being restricted to genes unique to each individual cluster.

### 4.6. Identification of Differentially Expressed Genes between WT and Ts65Dn

Within each cell type, WT and TS samples were compared for differential gene expression using Seurat’s FindMarkers function. To be included in the analysis, the gene had to be expressed in at least 10% of the cells from one of the two groups for that cell type and there had to be at least a 0.25-fold change in gene expression between genotypes. After correcting for multiple testing, only genes with a *p*-adjusted value < 0.001 were considered for downstream analyses.

### 4.7. Gene Set Enrichment

Using a hypergeometric test (shinyGO) [61], the differential expression signatures from each cellular subtype were examined for enriched Gene Ontology processes. Processes were classified as considerably enriched when their *p*-adjusted value was less than 0.05. The universe for the hypergeometric test was the entire list of genes found in the dataset.

### 4.8. Cellular Proportion

The proportional fraction of nuclei in each cell type was standardized to the total number of nuclei taken from each library to acquire insight into cell type variations in the trisomic hippocampus. We used single-cell differential composition analysis (scDC) to bootstrap proportion estimates for our samples to see if any changes in cell-type proportion were statistically significant [62]. 

### 4.9. Disease-Associated Microglia (DAM) Markers

To check for a DAM-associated transcriptional signature, we compared the DEGs of trisomic microglia with the DAM DEGs transcriptional signature of previously published datasets [37,38]. The significance of the overlap between both lists of genes was tested using the GeneOverlap package in R (v 1.38). 

### 4.10. Pseudotime Analysis

To infer the pseudotime of microglia in both conditions we used functions provided with the Monocle 2 package (version 2.6.4) [63,64]. The cell trajectory was defined based on the top 50 most differentially expressed genes between the two microglial clusters, corresponding to the two states of activation.

### 4.11. Statistical Analysis

When two conditions were compared, the Shapiro–Wilks test was conducted to check the normality of the data, and Fisher’s F test was used to assess the homogeneity of variances between groups. When data met the assumptions of parametric distribution, results were analyzed by unpaired student’s *t*-test. Paired *t*-tests were employed to compare paired variables. Mann–Whitney–Wilcoxon test was applied in cases where the data did not meet the requirements of normal distribution. Statistical analyses were two-tailed. The statistical test used is indicated in every Figure. Differences in means were considered statistically significant at *p* < 0.05.

Data analysis and statistics were performed using R studio (Version 1.1.463).

## Figures and Tables

**Figure 1 ijms-25-03289-f001:**
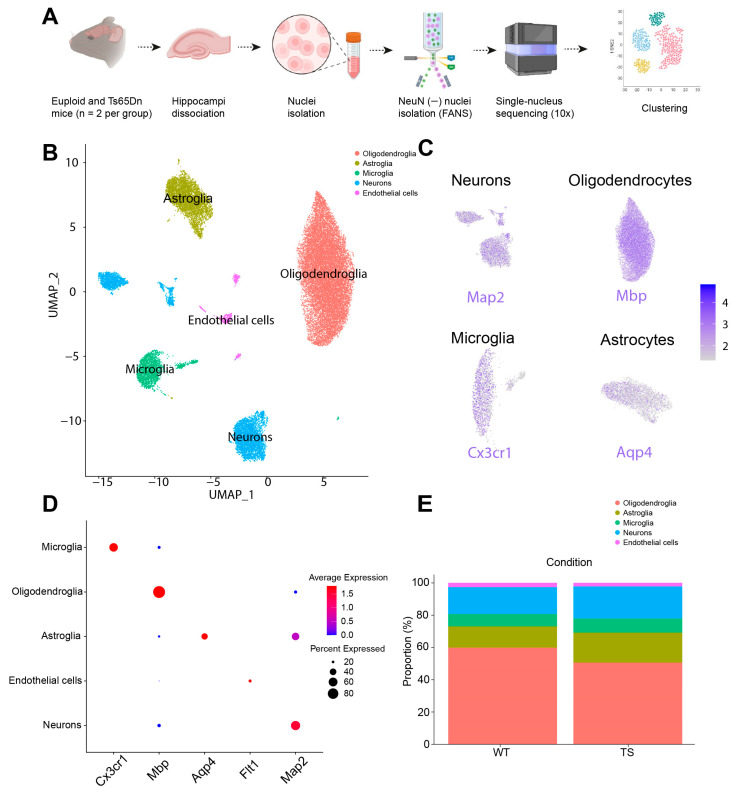
Single-nucleus sequencing and cell-type identification. (**A**) Overview of the experimental approach with a schematic showing the NeuN-negative cell sorting (see Section 4). Hippocampus of WT and Ts65Dn were collected and the nuclei suspension was prepared using enzymatic digestion and mechanical dissociation. Nuclei were incubated with anti-NeuN-Alexa647 and NeuN-negative nuclei were selected by FANS and subsequently sequenced using 10x sequencing. (**B**) All hippocampus single nuclei embedded in UMAP, displaying cell clusters in different colors. Each dot represents a single nucleus. (**C**) Mapping of microglia (*Cx3cr1*), astroglia (*Aqp4*) and oligodendroglia (*Mbp*) markers. (**D**) Dot plot showing enrichment of canonical markers of microglia, oligodendroglia, astroglia, endothelial cells, and neurons. (**E**) Proportion of the main cell populations both in WT and Ts65Dn hippocampus. FANS: fluorescent activated nuclear sorting.

**Figure 2 ijms-25-03289-f002:**
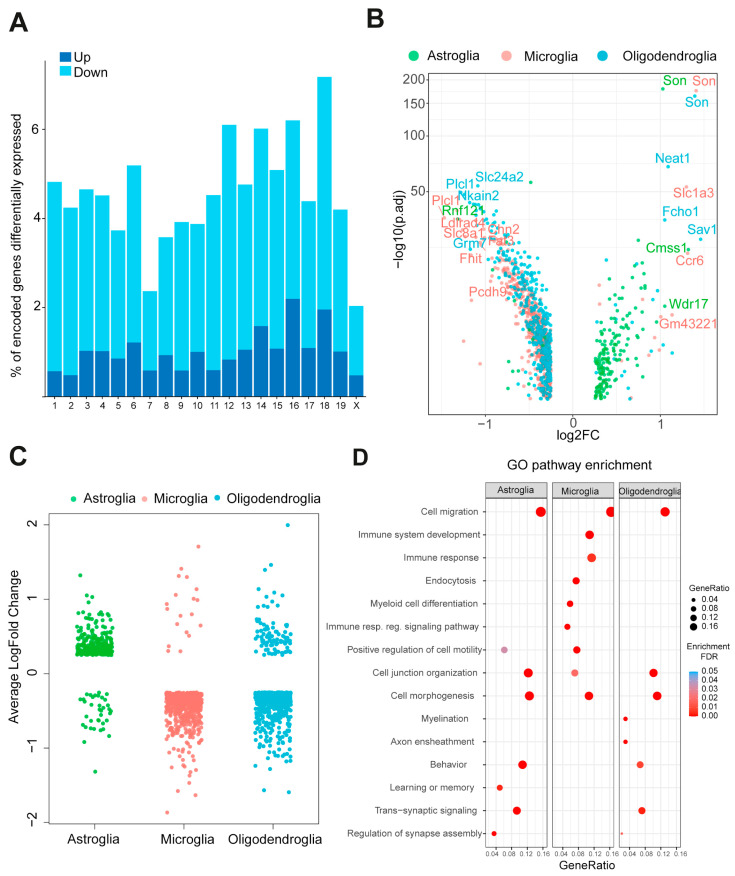
Transcriptionally different cell populations within the Ts65Dn hippocampus. (**A**) Distribution of DEGs along the mouse chromosomes. (**B**) Volcano plot of DEGs colored by each major glial type. (**C**) Strip plot showing the average log fold change of the different DEGs colored by glial cell type. (**D**) Biological pathways enriched for DEGs identified in each major glial cell type. DEGs differentially expressed genes.

**Figure 3 ijms-25-03289-f003:**
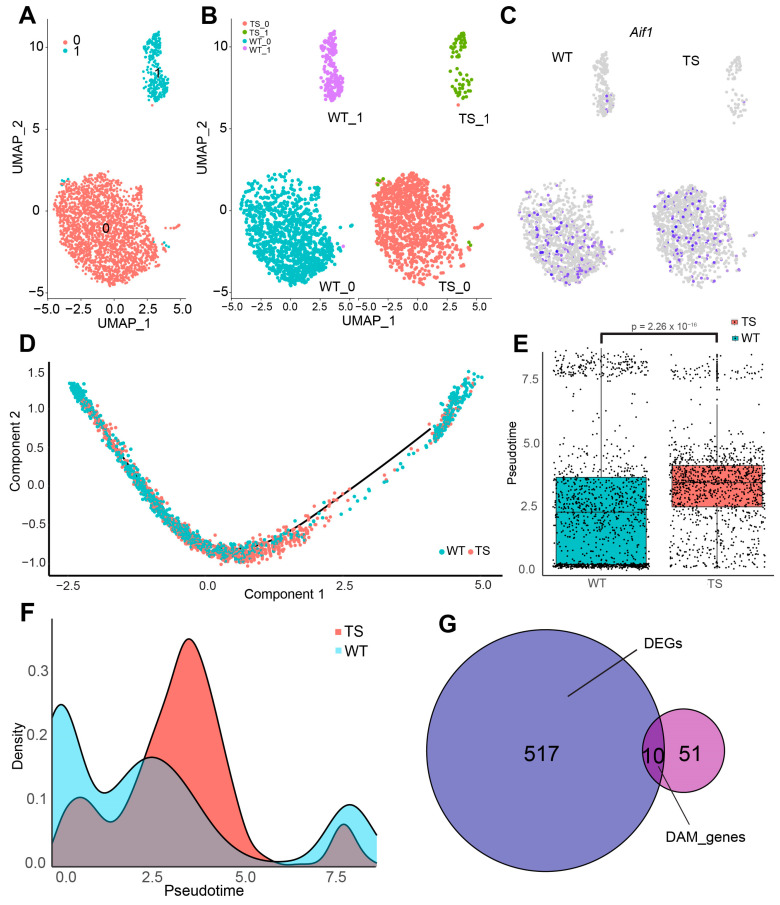
Transcriptional shift in trisomic microglial cells. (**A**) UMAP plot showing the microglia cluster both in WT (blue) and trisomic (red) mice. (**B**) UMAP plot showing the different clusters of microglia both in Ts65Dn (left) and WT mice (right). (**C**) Mapping of the *Aif* marker (encoding for Iba1) both in Ts65Dn (left) and WT (right) mice. (**D**) Pseudotime showing the relative positioning of cells along the trajectory quantifying the relative reactivity of microglial cells. (**E**) Box plot showing the pseudotime score of different microglial cells according to the top 50 DEG genes of both the homeostatic microglia cluster and reactive microglial cluster in Ts65Dn (red and WT mice (blue) mice. *p*-value = 2.26 × 10^−16^. (**F**) Relative abundance or density of microglial cells alongside the different pseudotime scores in both Ts65Dn (red) and WT (blue) hippocampus. (**G**) Venn diagram showing overlap between the DEGs in microglia and DAM signature. DEGs differentially expressed genes, DAM disease-associated microglia.

**Table 1 ijms-25-03289-t001:** DAM markers in trisomic microglia.

Genes	Avg_log2FC	Avg_log2FC Grn.KO	*p*-Value (Adjusted)
*Csmd3*	−1.8672	−1.148	1.73 × 10^−95^
*Cables1*	−0.4903	−0.447	7.69 × 10^−10^
*Ptprm*	−1.3229	−0.399	3.16 × 10^−40^
*Plxdc2*	−0.6931	−0.34	3.35 × 10^−15^
*Rapgef5*	−0.5133	−0.313	6.18 × 10^−15^
*Tanc2*	−0.79899	−0.252	1.61 × 10^−26^
*Bin1*	−0.33	−0.094	6.49 × 10^−8^
*Ldlrad4*	−1.2606	−0.022	1.19 × 10^−31^
*Fcrls*	0.7973	0.009	2.55 × 10^−13^
*Stab1*	0.5704	0.269	9.19 × 10^−4^

First column indicates the identified DAM genes in Ts65Dn microglia. Second column indicates the log fold change of the DAM genes in trisomic microglia. Third column indicates the log fold change of the genes described in previous studies (Grn.KO) [37]. Last column indicates the adjusted *p*-value of statistically significant DEGs that overlap with DAMs in different published datasets [37,38]. DAM disease-associated microglia.

## Data Availability

Data can be checked at: https://crgcnag-my.sharepoint.com/:f:/g/personal/csierra_crg_es/Esrwr5mSzk5IvvC2lopxRIMBOtUymZOwxpc1V2HjzVhKdg?e=cJbHIU (accessed on 11 February 2024).

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
