# Peer review of "Beyond Quiescent and Active: Intermediate Microglial Transcriptomic States in a Mouse Model of Down Syndrome"

_ijms, 2024, doi:10.3390/ijms25063289_

Round 1

Reviewer 1 Report

Comments and Suggestions for Authors

In this study, the authors used single nucleus RNA sequencing and investigated how trisomy affect microglial states in the Ts65Dn mouse model of DS. Based on the data analysis, they focused on microglia and identified differential expression of genes associated with different microglial states, including DAMs, ARMs, and human HAMs in trisomic microglia. One interesting finding through pseudotime analysis reveals a unique reactivity profile in Ts65Dn microglia, with fewer in a homeostatic state and more in an intermediate aberrantly reactive state than in euploid microglia.

Overall, the manuscript is clear, well written and organized and the data is interesting and well presented. However, I have several concerns about the data.

1.     The data showing that intermediate microglial transcriptomic states in a mouse model of Down syndrome, what about the corresponding functions of this kind state of microglia?

2.     Pseudotime analysis data in Figure 2D,E is one of highlights in this paper, how to identify the unique reactivity profile in Ts65Dn microglia? Better to confirm with staining.

3.     The authors identified differential expression of genes associated with different microglial states, including DAMs, ARMs, and human HAMs in trisomic microglia, it looks like no unique signatures of DS, better to add more discussion to analyze the difference.

Comments on the Quality of English Language

In this study, the authors used single nucleus RNA sequencing and investigated how trisomy affect microglial states in the Ts65Dn mouse model of DS. Based on the data analysis, they focused on microglia and identified differential expression of genes associated with different microglial states, including DAMs, ARMs, and human HAMs in trisomic microglia. One interesting finding through pseudotime analysis reveals a unique reactivity profile in Ts65Dn microglia, with fewer in a homeostatic state and more in an intermediate aberrantly reactive state than in euploid microglia.

Overall, the manuscript is clear, well written and organized and the data is interesting and well presented. However, I have several concerns about the data.

1.     The data showing that intermediate microglial transcriptomic states in a mouse model of Down syndrome, what about the corresponding functions of this kind state of microglia?

2.     Pseudotime analysis data in Figure 2D,E is one of highlights in this paper, how to identify the unique reactivity profile in Ts65Dn microglia? Better to confirm with staining.

3.     The authors identified differential expression of genes associated with different microglial states, including DAMs, ARMs, and human HAMs in trisomic microglia, it looks like no unique signatures of DS, better to add more discussion to analyze the difference.

Author Response

 The data showing that intermediate microglial transcriptomic states in a mouse model of Down syndrome, what about the corresponding functions of this kind state of microglia?

It is important to acknowledge that although the identification of these aberrant activation states in Ts65Dn microglia is novel in DS research, the implications of this discovery remain largely uncertain. Within the field of microglia research there is not a clear consensus about the implications of the wide gradation of microglial states, which adds complexity in order to interpret these results. 

Our results suggest that microglial response in DS disease may be more complex than previously anticipated, as evidenced by the discovery of intermediate microglial activation states in Ts65Dn mice. However, the functional significance of these intermediate states is still speculative. The microglial response to neuroinflammation and synaptic dysfunction associated with DS may exhibit transitional stages, which could be represented by intermediate activation levels. Alternatively, they might imply a unique activation phenotype with distinct functional outcomes. Some implications of this aberrant microglial states might include:

  1. Intermediate activation states might be linked to dysregulated immune responses, which might contribute to a more pronounced neuroinflammation and neuronal injury in DS.
  2. Aberrant microglial activation states may disrupt neuronal synaptic function by means of altered synaptic pruning mechanisms, possibly exacerbating cognitive impairments in DS. 
  3. Dysregulated microglial activation during critical periods of brain development in Ts65Dn mice might disrupt neuronal circuit formation, potentially contributing to part of the neurodevelopmental alterations observed in Ts65Dn mice.  

Even so, further investigations are needed to elucidate the implications of this aberrant state in DS pathology.

We have discussed it in the manuscript (lines 297 to 303).

  1. Pseudotime analysis data in Figure 2D,E is one of highlights in this paper, how to identify the unique reactivity profile in Ts65Dn microglia? Better to confirm with staining.

The reviewer is right that it would be great to confirm the distinct reactivity profile in Ts65Dn microglia with staining. Unfortunately, there are strong limitations for the use of immunofluorescence in identifying intermediate activation states of microglia. In contrast to the detailed gene expression profiling at the single-cell level, only a small number of markers are available for immunofluorescence staining, and those mainly classify cells into broad activation states (e.g., quiescent vs. activated). Although we recognize the need for validation studies and appreciate the recommendation, we feel that the combination of snRNA-seq data and pseudotime analysis provides enough evidence to validate our conclusions about the distinct reactivity profile of Ts65Dn microglia. Microglia in intermediate activation stages may exhibit subtle alterations in gene expression that are not easily captured by conventional immunofluorescence markers of microglial activation such as Iba1, Cd68 and Cd45, among others. Through transcriptome investigations like snRNA-seq, it is possible to more precisely examine the differential expression of genes linked to particular activation pathways, which may characterize these intermediate states. Pseudotime analysis also makes it possible to identify transitional states based on the temporal ordering of changes in gene expression and reconstruct biological trajectories. Static immunofluorescence labeling may not fully capture the dynamic nature of cellular responses and may not be appropriate for precisely detecting intermediate activation states of microglia. Therefore, we think that a more reliable and thorough method for describing the distinct reactivity profile of Ts65Dn microglia in this work is pseudotime analysis based on snRNA-seq data.

We have discussed it in lines 287 to 291 in the manuscript. 

  1.     The authors identified differential expression of genes associated with different microglial states, including DAMs, ARMs, and human HAMs in trisomic microglia, it looks like no unique signatures of DS, better to add more discussion to analyze the difference.

Thank you for your insightful comment regarding the identification of differential gene expression associated with various microglial states in trisomic microglia. We appreciate your suggestion to add further discussion to analyze the specific differences that may constitute unique signatures of Down syndrome in our data. Indeed, we have found patterns of differential expression in microglial states, we have not specifically discussed the distinct signatures of DS in these microglial states. We agree that this is an important aspect to consider, and have incorporated it in the discussion section 

We have included a discussion in the manuscript (lines 325 to 329).

Reviewer 2 Report

Comments and Suggestions for Authors

In this study the authors demonstrated that a different and aberrant state of trisomic microglia might account for a higher microglia reactivity that is reported in DS and in DS mouse.

The major aim of this work is, as reported by the authors, to investigate the role of trisomy on the microglial transcriptomic profile and if it can impact on differential microglial states within the hippocampus of a specific mouse model that recapitulates most of traits observed in DS (Ts65Dn).

This, I think, is a relevant topic that addresses the gap on the lacking of knowledge about the classification of microglia: M1 versus M2 or different intermediate states in DS, besides of determining the microglial transcriptomic profiles in hippocampus of Ts65Dn.

These findings have been reported in the paper: one issue that the authors could perform is to move the conclusions reported in the introduction section to the conclusions section (lines 313-317).

The topic is interesting as well as the results obtained, from what I understand, the methods are sound.

I have some minor points to address:

1. in the abstract section, there are two close repetitions: “however” line 19 and line 20.

2. In the aim of the work, from line 85, the authors included also the results and the conclusions. I think that the sentences from line 94 to line 109 should be deleted.

3. The authors should pay attention to acronyms particularly in the legend of the figures.

4. The authors should pay attention to English errors: for instance, line 259 “This finding concur…”

5. To make the discussion easier to read, the authors could divide it into subparagraphs, perhaps with titles following the structure of the results.

6. The authors could add a Limitations section.

7. In the Materials and Methods, the authors have included a reference not following the guidelines of the journal: Liu et al., 2003 line 323

Comments on the Quality of English Language

Minor editing 

Author Response

  1. In the abstract section, there are two close repetitions: “however” line 19 and line 20.

We have changed however in line 20 by “indeed”.

  1. In the aim of the work, from line 85, the authors included also the results and the conclusions. I think that the sentences from line 94 to line 109 should be deleted.

We have deleted from line 101 to line 112.

  1. The authors should pay attention to acronyms particularly in the legend of the figures.

We have added the acronyms in the legend of the figures, when appropriate.

  1. The authors should pay attention to English errors: for instance, line 259 “This finding concur…”

We have corrected it in the manuscript (line 273).

  1. To make the discussion easier to read, the authors could divide it into subparagraphs, perhaps with titles following the structure of the results.

We have divided the discussion into subparagraphs to make the discussion easier to read. 

  1. The authors could add a Limitations section.

We have added a limitations section (Section 4) after the discussion (lines 353 to 372).

  1. In the Materials and Methods, the authors have included a reference not following the guidelines of the journal: Liu et al., 2003 line 323

We have corrected it in the manuscript (line 379).

Round 2

Reviewer 1 Report

Comments and Suggestions for Authors

I have no more comments.